# Dual-Resolution Fusion Modeling for Unsupervised Cross-Resolution Person Re-Identification

Zhiqi Pang
Faculty of Computing
Harbin Institute of Technology
Harbin, China
22b903055@stu.hit.edu.cn

Lingling Zhao
Faculty of Computing
Harbin Institute of Technology
Harbin, China
zhaoll@hit.edu.cn

Chunyu Wang*
Faculty of Computing
Harbin Institute of Technology
Harbin, China
chunyu@hit.edu.cn

## Abstract

Cross-resolution person re-identification (CR-ReID) aims to match images of the same person with different resolutions in different scenarios. Existing CR-ReID methods achieve promising performance by relying on large-scale manually annotated identity labels. However, acquiring manual labels requires considerable human effort, greatly limiting the flexibility of existing CR-ReID methods. To address this issue, we propose a dual-resolution fusion modeling (DRFM) framework to tackle the CR-ReID problem in an unsupervised manner. Firstly, we design a cross-resolution pseudo-label generation (CPG) method, which initially clusters high-resolution images and then obtains reliable identity pseudo-labels by fusing class vectors in both resolution spaces. Subsequently, we develop a cross-resolution feature fusion (CRFF) module to fuse features from both high-resolution and low-resolution spaces. The fusion features have the potential to serve as a new form of resolution-invariant features. Finally, we introduce cross-resolution contrastive loss and probability sharpening loss in DRFM to facilitate resolution-invariant learning and effectively utilize ambiguous samples for optimization. Experimental results on multiple CR-ReID datasets demonstrate that the proposed DRFM not only outperforms existing unsupervised methods but also approaches the performance of early supervised methods.

## CCS Concepts

• **Information systems** → **Information retrieval**.

## Keywords

Unsupervised learning, Cross-resolution person re-identification, Feature fusion, Contrastive learning

**ACM Reference Format:**
Zhiqi Pang, Lingling Zhao, and Chunyu Wang. 2024. Dual-Resolution Fusion Modeling for Unsupervised Cross-Resolution Person Re-Identification. In *Proceedings of the 32nd ACM International Conference on Multimedia (MM '24), October 28-November 1, 2024, Melbourne, VIC, Australia.* ACM, New York, NY, USA, 10 pages. https://doi.org/10.1145/3664647.3681067

*Corresponding author.

## 1 Introduction

Person re-identification (ReID) [44, 45, 53] targets matching images of individuals captured by multiple cameras with the same identity. Benefiting from deep models [37, 42] and optimization methods [20, 36], existing ReID methods have achieved exciting performance in simple scenarios. However, their performance can be influenced by various factors in complex scenarios [4, 41, 52]. For instance, due to the impact of camera specifications and shooting distances, there is often a significant disparity in the underlying resolution (clarity) among different images. Matching low-resolution (LR) images directly with high-resolution (HR) images can lead to performance degradation. To address this issue, some researchers have turned their attention to cross-resolution person re-identification (CR-ReID) [41, 52]. Existing CR-ReID methods [16, 48] typically begin by utilizing super-resolution (SR) models [23, 50] to enhance the underlying resolution of LR images, thereby obtaining synthetic HR (SHR) images. Subsequently, matching is performed between SHR and HR images.

While existing CR-ReID methods have achieved promising performance, they still rely on manually annotated identity labels [41, 52]. However, annotating datasets for image retrieval tasks is a time-consuming process [33], and currently, there are no unsupervised CR-ReID methods available. Therefore, in this paper, we attempt to address the CR-ReID problem in an unsupervised manner. In this new scenario, we face two key challenges: (1) How to obtain reliable identity pseudo-labels. Unsupervised general ReID methods [27, 38] often use clustering algorithms [11] to generate pseudo-labels. However, in CR-ReID, due to significant appearance differences between LR and HR images, directly applying clustering algorithms between them can result in a large number of noisy labels. Noisy labels can mislead model optimization and hinder performance improvement [15, 32]. (2) How to obtain resolution-invariant features. On one hand, SHR images generated by SR models [23, 50] may contain artifacts, which could alter the original identity features. Therefore, using SHR features as resolution-invariant features for image matching is suboptimal. On the other hand, unsupervised methods often struggle to incorporate unlabeled ambiguous samples into the optimization process [15, 32], leading to lower sample utilization compared to supervised methods. This further increases the difficulty of resolution-invariant feature learning. For smooth and concise description, in the subsequent sections, we will also refer to the person images interchangeably as samples, based on which term fits best in the context.

In this paper, we propose a novel unsupervised CR-ReID method: dual-resolution fusion modeling (DRFM). As illustrated in Figure 1, on one hand, SHR images contain clearer details compared to LR

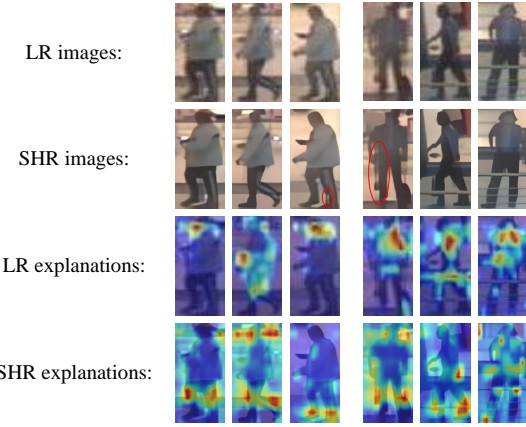

LR images:

SHR images:

LR explanations:

SHR explanations:

**Figure 1: Examples of LR images, SHR images, LR explanations, and SHR explanations. Artifacts in SHR images are highlighted with red circles.**

images but inevitably include some artifacts. On the other hand, the model exhibits significant differences in visual explanations [35] between LR and SHR images. Therefore, the complementary nature between SHR and LR images holds the potential to improve the performance of CR-ReID [52]. Unlike supervised CR-ReID [52], the unsupervised CR-ReID problem we face is more complex and requires deeper fusion. Therefore, DRFM fuses the semantic information of LR and SHR images at both the class and feature levels, thereby simultaneously addressing the aforementioned two challenges. To obtain reliable pseudo-labels, we introduce the cross-resolution pseudo-label generation (CPG) method for class-level fusion. CPG initially clusters HR images and passes pseudo-labels to synthetic LR (SLR) images. Then, it estimates class vectors for SHR and LR images based on the pseudo-labels from HR and SLR images, respectively. Finally, it fuses the class vectors from both spaces to assign pseudo-labels to LR and SHR images. To obtain resolution-invariant features, we first design the cross-resolution feature fusion (CRFF) module in DRFM. The fusion of features from two different spaces has the potential to serve as a novel form of resolution-invariant feature. Subsequently, we introduce the cross-resolution contrastive loss and probability sharpening loss to optimize the entire framework. The former aims to facilitate resolution-invariant feature learning, while the latter aims to further enhance model performance by fully utilizing unlabeled ambiguous samples.

The main contributions are summarized as follows:

- We propose a dual-resolution fusion modeling (DRFM) framework, which integrates semantic information at both the class and feature levels to address the unsupervised CR-ReID problem. To the best of our knowledge, this is the first attempt in the field of unsupervised CR-ReID.
- We design a cross-resolution pseudo-label generation (CPG) method, which acquires reliable pseudo-labels by fusing class vectors of LR and SHR images, instead of directly applying clustering algorithms between HR and LR images.

- We develop a cross-resolution feature fusion (CRFF) module to obtain resolution-invariant features. Additionally, we introduce a cross-resolution contrastive loss and probability sharpening loss to facilitate resolution-invariant feature learning.
- Extensive experimental results on multiple datasets demonstrate that our method not only surpasses existing unsupervised methods but also approaches the performance of certain supervised methods.

## 2 Related Work

### 2.1 Unsupervised general ReID

Existing unsupervised general ReID methods can be roughly divided into unsupervised domain adaptation (UDA) methods [21, 26, 31] and fully unsupervised (FU) methods [1, 30, 46]. The former relies on labeled source domain data and unlabeled target domain data for training, aiming to improve the recognition performance on the target domain. The latter no longer rely on labeled source domain data, making them more flexible and challenging. FU method typically iterates between sample clustering and model optimization. For instance, in earlier approaches, BUC [30] utilizes a hierarchical clustering algorithm to generate pseudo-labels, followed by the introduction of softmax classification loss to optimize the model. Building upon BUC [30], HCT [47] incorporates batch hard triplet loss [20] to address hard samples.

Subsequent research often focuses on addressing camera gap [34, 43] and noisy labels [7, 49]. For instance, during the clustering stage, IICS [43] amplifies inter-camera similarity by concatenating scores of the same image across different classifiers. CIFL [34] achieves a similar objective by introducing the concept of ensemble learning. In the optimization stage, both ICE [1] and O2CAP [39] introduce optimization methods tailored to camera gap to align distributions across cameras. To address noisy labels, RLCC [49] refines pseudo-labels using the concept of temporal ensembling. Both PPLR [7] and SECRET [18] leverage the complementary relationship between global and local features to eliminate noisy labels. RMCL [32] evaluates the reliability of pseudo-labels from the perspectives of certainty and stability, assigning lower weights to noisy labels. To mitigate the impact of noisy labels, Purification [25] initially trains a teacher model from the original pseudo-labels. The teacher model is then employed to guide the learning of a student model. The student model can converge rapidly under the supervision of the teacher model, thereby reducing the interference of noisy labels. While the aforementioned methods have shown promising performance in simple scenarios, they often face significant performance degradation in cross-resolution settings.

### 2.2 Cross-resolution ReID

CR-ReID aims to address the issue of resolution mismatch between images [6, 22]. Existing CR-ReID methods typically employ SR models [23, 50] to enhance the LR images' underlying resolution, thereby restoring missing fine-grained information. For instance, SING [22] performs joint training by connecting an SR model and a ReID model, facilitating representation learning while generating high-resolution images. PRI [16] incorporates a scale predictor module designed for the SR model, thereby providing appropriate

scales for the super-resolution process. INTACT [6] utilizes the underlying association knowledge between SR and ReID as an additional learning constraint, thereby enhancing the compatibility between SR and ReID models. While SR models have the potential to restore missing fine-grained information in LR images, they inevitably introduce artifacts. These artifacts often interfere with the extraction of identity-related features.

In addition to utilizing SR models for cross-resolution alignment, Chen et al. [3] introduce the concept of adversarial training [13] during the representation learning process, aiming to align the feature distributions of LR and HR images. Similarly, CAD-Net [29] also incorporates adversarial training to obtain resolution-invariant features. However, since HR images contain fine-grained discriminative information not present in LR images, directly aligning the feature distributions of HR and LR images often results in the loss of such information. To address this issue, LRAR [41] embeds resolution information into features to obtain varying-length representations. The HR features are composed of shared components and HR-specific components, thereby preserving fine-grained information exclusive to HR while learning resolution-invariant feature representations. While the aforementioned methods have shown promising performance, they all rely on manually annotated identity labels. In contrast to these approaches, we propose an unsupervised CR-ReID method with the aim of reducing the dependence on manually annotated labels.

## 3 Proposed Framework

### 3.1 Overview

During the training phase, we are provided with a set of HR images $\{x_i^h\}_{i=1}^{N_1}$ and a set of LR images $\{x_i^l\}_{i=1}^{N_2}$, where $N_1$ and $N_2$ represent the numbers of HR and LR images, respectively. Our objective is to optimize the CR-ReID model without accessing manually annotated identity labels. During the testing phase, relying on the feature extraction capability of the model, we search for images in the HR image gallery that share the same identity as a given LR query image based on feature similarity.

To achieve the aforementioned objectives, we propose the dual-resolution fusion modeling (DRFM) framework. As illustrated in Figure 2, in the preprocessing (Preproc) stage, DRFM performs down-sampling (DS) on $\{x_i^h\}_{i=1}^{N_1}$ and super-resolution (SR) on $\{x_i^l\}_{i=1}^{N_2}$ to generate SLR image set $\{x_i^{sl}\}_{i=1}^{N_1}$ and SHR image set $\{x_i^{sh}\}_{i=1}^{N_2}$, respectively. In the optimization stage, DRFM initially employs the HR encoder $E_h$ to extract features from $\{x_i^h\}_{i=1}^{N_1}$ and $\{x_i^{sh}\}_{i=1}^{N_2}$, obtaining HR features $\{f_i^h\}_{i=1}^{N_1}$ and SHR features $\{f_i^{sh}\}_{i=1}^{N_2}$. It then utilizes the LR encoder $E_l$ to extract features from $\{x_i^l\}_{i=1}^{N_2}$ and $\{x_i^{sl}\}_{i=1}^{N_1}$, resulting in LR features $\{f_i^l\}_{i=1}^{N_2}$ and SLR features $\{f_i^{sl}\}_{i=1}^{N_1}$. The predictor $Pred$ is used to map SHR features. Subsequently, DRFM utilizes the cross-resolution pseudo-label generation (CPG) to obtain pseudo-labels, and employs the cross-resolution feature fusion (CRFF) module to generate fusion features $\{f_i^{hsl}\}_{i=1}^{N_1}$ and $\{f_i^{lsh}\}_{i=1}^{N_2}$. It utilizes memory $M_{hh} = \{v_i^{hh}\}_{i=1}^{N}$, memory $M_{ll} = \{v_i^{ll}\}_{i=1}^{N}$, and memory $M_{fu} = \{v_i^{fu}\}_{i=1}^{N}$ to store features with pseudo-labels in the HR space, LR space, and fusion space, respectively. Here, $N$ ($N < N_1 + N_2$) represents the number of features with pseudo-labels.

**Table 1: Key notations used in the paper.**

| Notation | Meaning |
|---|---|
| $f_i^h$ | the feature of HR image $x_i^h$ |
| $f_i^l$ | the feature of LR image $x_i^l$ |
| $f_i^{sh}$ | the feature of SHR image $x_i^{sh}$ |
| $f_i^{sl}$ | the feature of SLR image $x_i^{sl}$ |
| $f_i^{hsl}$ | the fusion feature of $f_i^h$ and $f_i^{sl}$ |
| $f_i^{lsh}$ | the fusion feature of $f_i^l$ and $f_i^{sh}$ |
| $f_i^{hh}$ | the collective term for $f_i^h$ and $f_i^{sh}$ with pseudo-labels |
| $f_i^{ll}$ | the collective term for $f_i^l$ and $f_i^{sl}$ with pseudo-labels |
| $f_i^{fu}$ | the collective term for $f_i^{hsl}$ and $f_i^{lsh}$ with pseudo-labels |
| $v_i^{hsl}$ | the offline $f_i^{hsl}$ stored in $M_{fu}$ |
| $v_i^{lsh}$ | the offline $f_i^{lsh}$ stored in $M_{fu}$ |
| $v_i^{fu}$ | the collective term for $v_i^{hsl}$ and $v_i^{lsh}$ |

Finally, DRFM optimizes using cross-resolution contrastive loss $L_{cc}$, probability sharpening loss $L_{ps}$, and identity consistency loss $L_{ic}$.

In the optimization stage, we update the memory separately:

$$v_i^{hh} \leftarrow \alpha v_i^{hh} + (1-\alpha) f_i^{hh}, \tag{1}$$

$$v_i^{ll} \leftarrow \alpha v_i^{ll} + (1-\alpha) f_i^{ll}, \tag{2}$$

$$v_i^{fu} \leftarrow \alpha v_i^{fu} + (1-\alpha) f_i^{fu}, \tag{3}$$

where $f_i^{hh}, f_i^{ll}$, and $f_i^{fu}$ represent features with pseudo-labels in the HR space, LR space, and fusion space, respectively, which are output by the encoder or CRFF. $\{f_i^{hh}\}_{i=1}^{N} \subseteq \{f_i^h\}_{i=1}^{N_1} \cup \{f_i^{sh}\}_{i=1}^{N_2}$, $\{f_i^{ll}\}_{i=1}^{N} \subseteq \{f_i^l\}_{i=1}^{N_2} \cup \{f_i^{sl}\}_{i=1}^{N_1}$, $\{f_i^{fu}\}_{i=1}^{N} \subseteq \{f_i^{hsl}\}_{i=1}^{N_1} \cup \{f_i^{lsh}\}_{i=1}^{N_2}$, and $\alpha$ is the update rate. As shown in Table 1, for ease of understanding, we have provided a summary and explanation of the key notations used in this paper.

### 3.2 Cross-resolution pseudo-label generation

In cross-resolution scenarios, the resolution disparity between LR and HR images significantly increases feature discrepancies. Directly applying clustering algorithms between them can result in a large number of noisy labels. Intuitively, it seems reasonable to reduce the resolution disparity between images before applying clustering algorithms. However, this may not be straightforward. On one hand, down-sampling HR images can mitigate resolution disparity but results in the loss of fine-grained discriminative information. On the other hand, super-resolution generation for LR images can also reduce resolution disparity but inevitably introduces artifacts, which may alter the original identity features.

To address the aforementioned issues, we have designed the cross-resolution pseudo-label generation (CPG) method, which aims to integrate semantic information from both HR and LR spaces to generate reliable pseudo-labels. Specifically, since HR images contain unique and reliable fine-grained discriminative information, CPG initially performs clustering on HR images to generate pseudo-labels. For any $x_i^{sl}$, CPG assigns it the same pseudo-label as its corresponding $x_i^h$. Subsequently, CPG estimates class vectors

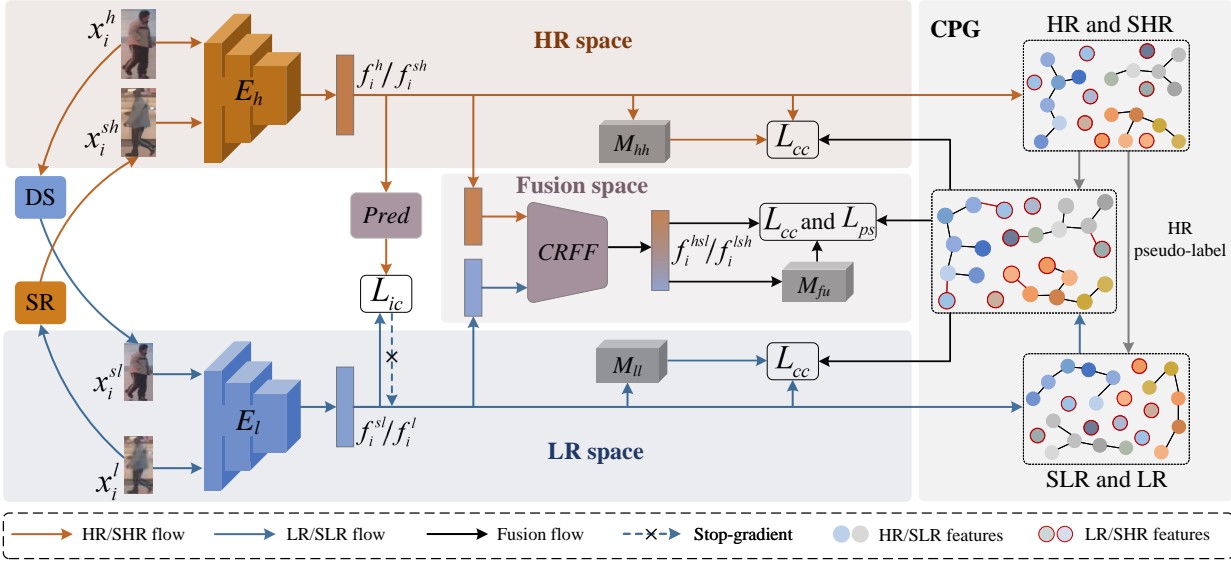

Figure 2: The overview of our proposed DRFM framework. DRFM comprises four trainable modules: the HR encoder $E_h$, LR encoder $E_l$, predictor $Pred$, and CRFF module. Memory $M_{hh}$, memory $M_{ll}$, and memory $M_{fu}$ store features with pseudo-labels in the HR space, LR space, and fusion space, respectively. $L_{cc}$, $L_{ps}$, and $L_{ic}$ are loss functions. In CPG, the connecting lines between features indicate that the connected features share the same pseudo-label. The black connecting lines are generated by clustering algorithm, while the red connecting lines are generated by the fusion of class vectors.

for SHR and LR images separately in both HR and LR spaces. For example, in LR space, the class vector of LR image $x_i^l$ is defined as:

$$s_i^l = \text{Softmax}(f_i^l \cdot C_{sl}^T / \tau_l), \tag{4}$$

where $f_i^l$ represents the feature extracted by encoder $E_l$ from $x_i^l$, $\tau_l$ is the temperature hyperparameter, and $C_{sl}$ is the matrix composed of SLR centroids. For example, the $j$-th SLR centroid $c_j^{sl}$ in $C_{sl}$ is defined as:

$$c_j^{sl} = \frac{1}{n_j^{sl}} \sum_{i=1}^{n_j^{sl}} f_i^{sl}, \tag{5}$$

where $f_i^{sl}$ represents the feature extracted by encoder $E_l$ from $x_i^{sl}$, and $n_j^{sl}$ is the number of SLR features in the $j$-th cluster. Similarly, the class vector of SHR image $x_i^{sh}$ is defined as:

$$s_i^{sh} = \text{Softmax}(f_i^{sh} \cdot C_h^T / \tau_h), \tag{6}$$

where $f_i^{sh}$ represents the feature extracted by encoder $E_h$ from $x_i^{sh}$, $\tau_h$ is the temperature hyperparameter, and $C_h$ is the matrix composed of HR centroids. For example, the $j$-th HR centroid $c_j^h$ in $C_h$ is defined as:

$$c_j^h = \frac{1}{n_j^h} \sum_{i=1}^{n_j^h} f_i^h, \tag{7}$$

where $f_i^h$ represents the feature extracted by encoder $E_h$ from $x_i^h$, and $n_j^h$ is the number of HR features in the $j$-th cluster. As $x_i^l$ and $x_i^{sh}$ share the same identity information, we calculate the fused

class vectors for them:

$$s_i^{lsh} = \frac{s_i^l \odot s_i^{sh}}{||s_i^l \odot s_i^{sh}||_1}, \tag{8}$$

where $|| \cdot ||_1$ represents $\ell_1-$norm, and $\odot$ denotes the Hadamard product. Subsequently, we focus on the maximum term $\max(s_i^{lsh})$ within $s_i^{lsh}$. When $\max(s_i^{lsh})$ is relatively large, it indicates that $s_i^l$ and $s_i^{sh}$ both exhibit high probability values for a certain class. The model typically has high confidence in the class corresponding to $\max(s_i^{lsh})$, meaning that samples $x_i^l$ and $x_i^{sh}$ are high-confidence samples for the model. Conversely, when $\max(s_i^{lsh})$ is small, it suggests that $s_i^l$ and $s_i^{sh}$ show high probability values for different classes, indicating that samples $x_i^l$ and $x_i^{sh}$ are ambiguous samples for the model. Therefore, based on $\max(s_i^{lsh})$, we independently sort SHR images and LR images in descending order and assign pseudo-labels corresponding to $\max(s_i^{lsh})$ to images in the top $T$ proportion of each sequence. Here, $T$ is defined as:

$$T = \frac{N_1^c}{N_1} \cdot t, \tag{9}$$

where $N_1^c$ represents the number of HR images assigned pseudo-labels by the clustering algorithm in the current stage, and $t$ is the proportional hyperparameter. We consider images with the same pseudo-labels in each resolution space as belonging to the same cluster.

The above process reveals that CPG possesses dual advantages. On one hand, CPG achieves semantic complementarity by fusing class vectors from both spaces, thereby obtaining reliable pseudo-labels. On the other hand, CPG provides abundant cross-resolution

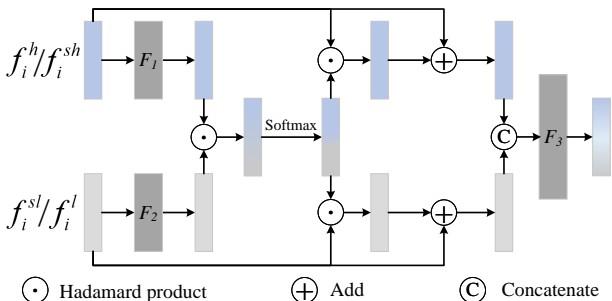

**Figure 3: Illustration of the cross-resolution feature fusion module.**

positive pairs for model optimization by forcibly grouping SHR images and LR images into HR and SLR clusters, respectively.

## 3.3 Cross-resolution feature fusion

Existing methods [6, 22] typically focus on aligning the SHR features with HR features to learn resolution-invariant feature representations. However, the SHR images generated by SR models [23, 50] may contain artifacts, which could alter the original identity features. Therefore, considering SHR features as resolution-invariant features for image matching is suboptimal. In response to the aforementioned issue, we have developed a cross-resolution feature fusion (CRFF) module to achieve the complementary nature between SHR and LR images. Subsequent matching will be based on the fusion features generated by CRFF. The detailed structure of CRFF is illustrated in Figure 3.

We illustrate the detailed processing steps of the CRFF module using SHR feature $f_i^{sh}$ and LR feature $f_i^l$ as examples. Similar procedures are followed for HR feature $f_i^h$ and SLR feature $f_i^{sl}$. We first calculate the similarity between $f_i^{sh}$ and $f_i^l$:

$$\tilde{f}_i^{lsh} = \text{Softmax}[F_1(f_i^{sh}) \odot F_2(f_i^l)], \tag{10}$$

where $F_1$ and $F_2$ represent fully connected (FC) layers, and $\odot$ denotes the Hadamard product. Subsequently, we perform feature enhancement based on similarity:

$$\hat{f}_i^{sh} = f_i^{sh} + f_i^{sh} \odot \tilde{f}_i^{lsh}, \tag{11}$$

$$\hat{f}_i^l = f_i^l + f_i^l \odot \tilde{f}_i^{lsh}, \tag{12}$$

Finally, feature fusion is accomplished based on feature concatenation (Cat) and FC layer $F_3$:

$$f_i^{lsh} = F_3[\text{Cat}(\hat{f}_i^{sh}, \hat{f}_i^l)], \tag{13}$$

$f_i^{lsh}$ shares the same pseudo-label with $f_i^l$, and similarly, $f_i^{hsl}$ shares the same pseudo-label with $f_i^h$. We introduce memory $M_{hh}$, memory $M_{ll}$, and memory $M_{fu}$ to store features with pseudo-labels in the HR space, LR space, and fusion space, respectively.

From the above process, it is evident that since $f_i^{lsh}$ and $f_i^{hsl}$ achieve semantic complementarity between two spaces, they have the potential to become a new form of resolution-invariant feature representation.

## 3.4 Resolution-invariant feature learning

During the optimization stage, existing supervised CR-ReID methods [6, 16, 22] typically utilize identity classification loss to guide model optimization. While this encourages the model to extract identity-related features, it does not prioritize resolution-invariance learning. Consequently, we have developed a cross-resolution contrastive loss aiming to simultaneously enhance identity-relevant and resolution-invariant feature learning. For simplicity, we illustrate this using the example of the fusion space. Based on the pseudo-labels provided by CPG and the offline features stored in memory $M_{fu} = \{v_i^{fu}\}_{i=1}^N$, we initially compute resolution centroids. For example, the resolution centroid $c_j^{hsl}$ is defined as:

$$c_j^{hsl} = \frac{1}{n_j^{hsl}} \sum_{i=1}^{n_j^{hsl}} v_i^{hsl}, \tag{14}$$

where $v_i^{hsl}$ represents the fusion feature of HR and SLR features stored in $M_{fu}$, and $n_j^{hsl}$ represents the number of $v_i^{hsl}$ in the $j$-th cluster. Similarly, the resolution centroid $c_k^{lsh}$ is defined as:

$$c_k^{lsh} = \frac{1}{n_k^{lsh}} \sum_{i=1}^{n_k^{lsh}} v_i^{lsh}, \tag{15}$$

where $v_i^{lsh}$ represents the fusion feature of LR and SHR features stored in $M_{fu}$, and $n_k^{lsh}$ represents the number of $v_i^{lsh}$ in the $k$-th cluster. It is possible for a cluster to contain two resolution centroids simultaneously. For any feature $f_i^{fu}$ with pseudo-label in the fusion space, we define the set of all resolution centroids in its cluster as its positive centroid set $P_i^{fu}$, and we define the set of $|Q|$ nearest resolution centroids in other clusters to $f_i^{fu}$ as its negative centroid set $Q_i^{fu}$. We increase the similarity of $f_i^{fu}$ to the positive centroid and decrease the similarity of $f_i^{fu}$ to the negative centroid:

$$L_{cc}(f_i^{fu}, P_i^{fu}, Q_i^{fu}) = -\frac{1}{|P_i^{fu}|} \sum_{c_p \in P_i^{fu}} \log \frac{\exp(f_i^{fu} \cdot c_p^T / \tau_c)}{\sum_{c_q \in P_i^{fu} \cup Q_i^{fu}} \exp(f_i \cdot c_q^T / \tau_c)}, \tag{16}$$

where $|P_i^{fu}| \in \{1, 2\}$ represents the number of centroids in set $P_i^{fu}$, and $\tau_c$ is the temperature hyperparameter. We perform a similar optimization process for the HR space and LR space, where the cross-resolution contrastive loss is defined as:

$$L_{cc} = L_{cc}(f_i^{fu}, P_i^{fu}, Q_i^{fu}) + L_{cc}(f_i^{hh}, P_i^{hh}, Q_i^{hh}) + L_{cc}(f_i^{ll}, P_i^{ll}, Q_i^{ll}), \tag{17}$$

where $L_{cc}(f_i^{hh}, P_i^{hh}, Q_i^{hh})$ and $L_{cc}(f_i^{ll}, P_i^{ll}, Q_i^{ll})$ represent the objective functions for the HR space and LR space, respectively.

To encourage the HR encoder to focus on identity-relevant information in SHR images rather than artifacts, we introduce the identity consistency loss:

$$L_{ic} = -\frac{Pred(f_i^{sh})}{||Pred(f_i^{sh})||_2} \cdot \text{Stopgrad}\left(\frac{f_i^l}{||f_i^l||_2}\right), \tag{18}$$

where $|| \cdot ||_2$ represents $\ell_2-$norm, $Pred(\cdot)$ represents the processing of the predictor, and Stopgrad$(\cdot)$ denotes the stop-gradient operation. The LR encoder cannot receive gradients from $L_{ic}$, which has been proven to effectively prevent model collapse [2].

While CPG can cluster high-confidence samples into existing clusters based on the fusion of class vectors, it inevitably reduces the utilization of ambiguous samples. To fully leverage ambiguous samples for model optimization, we have designed a probability sharpening loss. The distinction between high-confidence samples and ambiguous samples lies in the fact that the former exhibits extremely high probability values in only one class, while the latter typically shows relatively high probability values in several classes. However, they share a common characteristic of displaying low probability values in most classes. We refer to these classes with low probability values as reliable negative classes. In other words, while we cannot determine the unique positive class for ambiguous samples, we can confidently identify the majority of negative classes with high confidence. Therefore, the probability sharpening loss aims to reduce the predicted probabilities of ambiguous samples in the negative classes:

$$L_{ps} = -\text{Stopgrad}(\text{Softmax}(f_i^{lsh} \cdot C_{hsl}^T / \tau_s)) \cdot \log(\text{Softmax}(f_i^{lsh} \cdot C_{hsl}^T / \tau_c)), \quad (19)$$

where $\tau_s$ and $\tau_c$ are temperature hyperparameters, with $\tau_s < \tau_c$, and $C_{hsl}$ is the matrix composed of centroids $c_j^{hsl}$.

In summary, the overall objective function of DRFM is defined as:

$$L_{total} = L_{cc} + L_{ic} + \lambda L_{ps}, \quad (20)$$

where $\lambda$ is the weight coefficient of $L_{ps}$. The aforementioned optimization methods promote identity-relevant and resolution-invariant feature learning, forming a mutually reinforcing solution with CPG.

## 4 Experiment

### 4.1 Datasets and Evaluation Metrics

We evaluate existing methods and the proposed method on three multiple low-resolution (MLR) datasets. The details of each dataset are described as follows.

CAVIAR [5] is a real CR-ReID dataset, consisting of 1,220 images from 72 identities captured by two cameras. The resolution of the images captured by one camera is much lower than that of the images captured by the other camera. Following existing works [6, 29], we only use images from 50 identities. We split the dataset in half, utilizing images from 25 identities for training and images from the remaining 25 identities for testing.

MLR-CUHK03 is a synthetic CR-ReID dataset based on CUHK03 [28], containing images from 1,467 identities. Following existing works [6, 29], we use the benchmarking 1,367/100 training/test identity split. Both manually cropped and automatically detected images are used in our evaluations. To simulate resolution changes, we down-sample the resolution of images captured by one camera based on a random down-sampling rate $r \in \{2, 3, 4\}$, while the resolution of images captured by the other camera remains unchanged.

MLR-Market-1501 is a synthetic CR-ReID dataset based on Market-1501 [51]. It contains 32,668 images from 1,501 identities captured by six cameras, with 751 identities for training and 750 for testing. Following existing works [6, 29], we process the Market-1501 dataset in a similar manner to MLR-CUHK03 to obtain MLR-Market-1501.

During the testing phase, we construct the query set with all LR images per person, and the gallery set with one randomly selected HR image per person. We use cumulative matching characteristic

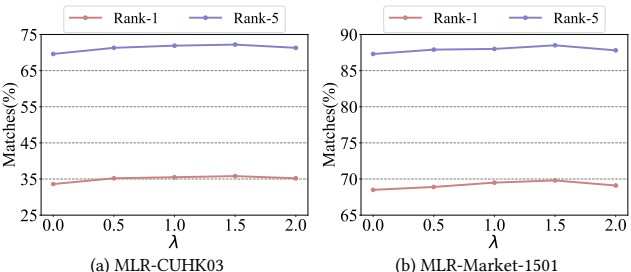

(a) MLR-CUHK03      (b) MLR-Market-1501

**Figure 4: Impact of hyperparameter $\lambda$ on performance.**

(CMC) to evaluate model performance and report Rank-1, Rank-5, and Rank-10.

### 4.2 Implementation Details

In the data preprocessing stage, we employ Swin2SR [8] to perform super-resolution (SR) on LR images to obtain SHR images. Additionally, we perform down-sampling (DS) on HR images to acquire SLR images. The scaling factors for both SR and DS are set to 4. Both random horizontal flipping and random cropping are adopted for data augmentation. We employ DBSCAN [11] for clustering the samples, and the minimum number of samples is set to 4. In DRFM, both encoders $E_h$ and $E_l$ utilize ResNet-50 [17], pre-trained on ImageNet [10]. The $Pred$, $F_1$ and $F_2$ all use a 2048×2048 full connection layer, and $F_3$ use a 4096×2048 full connection layer. For the temperature hyperparameters, we set $\tau_c$=0.04, $\tau_l$=0.04, $\tau_h$=0.05, and $\tau_s$=0.02. We optimize the DRFM through Adam optimizer [24] with a weight decay of 0.0005 and train the network with 40 epochs in total. The update rate $\alpha$ is set to 0.2 and the learning rate is set to 0.00035. For $L_{cc}$ and $L_{ic}$, we set the batch size to 64. For $L_{ps}$, the batch size is set to 16. For $L_{cc}$, we set $|Q| = 20$. In the initial training phase, only $L_{cc}$ and $L_{ic}$ are introduced. After the 10th epoch, $L_{ps}$ starts being used for model optimization.

### 4.3 Parameter Analysis

In this section, we conduct a detailed analysis of the key hyperparameters of DRFM on MLR-CUHK03 and MLR-Market-1501, including $\lambda$ and $t$.

*4.3.1 $\lambda$ of $L_{ps}$.* Figure 4 shows plots of the performance on the two datasets as function of the hyperparameter $\lambda$. Note that $\lambda = 0.0$ corresponds to the situation where the probability sharpening loss has no contribution to the overall training loss. We find that on MLR-CUHK03 and MLR-Market-1501, the model obtains the best performance when $\lambda = 1.5$. The results verify the generalization of the hyperparameter. The worst performance is achieved when $\lambda = 0.0$, which preliminarily verifies the effectiveness of $L_{ps}$.

*4.3.2 $t$ of CPG.* In Figure 5, we explore the optimal value of $t$ for CPG. On MLR-CUHK03, the model achieves the best performance when $t = 0.7$. On MLR-Market-1501, the model achieves the best performance when $t = 0.8$. When $t$ is too large or too small, the performance is poor. This is because when $t$ takes a small value, only a small number of samples are used for training, reducing the contribution of $L_{cc}$ to optimization. On the other hand, when $t$

**Table 2: Comparison of the proposed method with state-of-the-art methods on CAVIAR, MLR-CUHK03 and MLR-Market-1501. In the unsupervised setting, the top two performances are highlighted with bold and underline, respectively.**

| | Method | Reference | CAVIAR | | | MLR-CUHK03 | | | MLR-Market-1501 | | |
|---|---|---|---|---|---|---|---|---|---|---|---|
| | | | Rank-1 | Rank-5 | Rank-10 | Rank-1 | Rank-5 | Rank-10 | Rank-1 | Rank-5 | Rank-10 |
| Unsupervised | SpCL | NeurIPS20 | 10.4 | 33.6 | 54.6 | 25.6 | 58.9 | 75.1 | 56.7 | 80.6 | 87.9 |
| | ICE | ICCV21 | 9.8 | 32.1 | 55.1 | 27.9 | _67.1_ | _81.0_ | 62.4 | 84.4 | 89.9 |
| | CC | ACCV22 | 10.0 | 33.5 | 55.0 | _32.0_ | _67.1_ | 80.9 | 62.7 | 84.3 | 89.9 |
| | PPLR | CVPR22 | 8.3 | 31.6 | 53.6 | 23.7 | 56.0 | 72.9 | 61.9 | 84.3 | 90.0 |
| | Purification | TIP23 | _11.6_ | _38.8_ | 58.5 | 27.3 | 61.9 | 77.0 | _63.0_ | _84.7_ | _90.3_ |
| | DCCC | arXiv23 | 10.1 | 36.6 | _59.1_ | 24.2 | 55.7 | 72.0 | 56.2 | 79.1 | 86.2 |
| | DRFM | Ours | **12.8** | **39.1** | **59.9** | **35.8** | **72.2** | **83.6** | **69.8** | **88.5** | **92.5** |
| Supervised | SING | AAAI18 | 33.5 | 72.7 | 89.0 | 67.7 | 90.7 | 97.7 | 74.4 | 87.8 | 91.6 |
| | CSR-GAN | IJCAI18 | 34.7 | 72.5 | 87.4 | 71.3 | 92.1 | 97.4 | 76.4 | 88.5 | 91.9 |
| | CAD-Net | ICCV19 | 42.8 | 76.2 | 91.5 | 82.1 | 97.4 | 98.8 | 83.7 | 92.7 | 95.8 |
| | INTACT | CVPR20 | 44.0 | 81.8 | 93.9 | 86.4 | 97.4 | 98.5 | 88.1 | 95.0 | 96.9 |
| | PRI | ECCV20 | 43.2 | 78.5 | 91.9 | 85.2 | 97.5 | 98.8 | 84.9 | 93.5 | 96.1 |
| | PS-HRNet | TIP21 | - | - | - | 92.6 | 98.3 | 99.4 | 91.5 | 96.7 | 97.9 |
| | JBIM | IJCV22 | 52.0 | 83.1 | 94.4 | 88.3 | 97.2 | 98.7 | 88.1 | 95.1 | 96.9 |
| | LRAR | TIP23 | 63.6 | 79.2 | 96.6 | 89.2 | 98.9 | 99.8 | 90.1 | 96.2 | 97.7 |

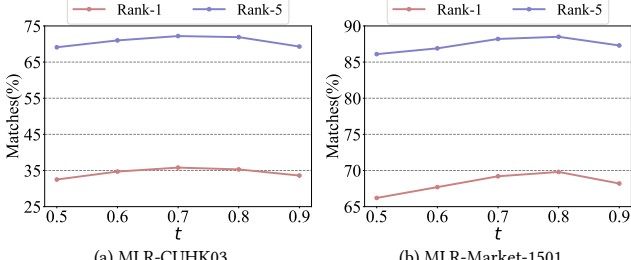

(a) MLR-CUHK03  (b) MLR-Market-1501

**Figure 5: Impact of hyperparameter $t$ on performance.**

takes a large value, too many noisy samples are used for training, misleading the optimization direction of the model.

## 4.4 Comparison with State-of-the-Arts

In Table 2, we compare DRFM with the state-of-the-art methods on CAVIAR, MLR-CUHK03 and MLR-Market-1501.

As there are no existing unsupervised CR-ReID methods, we reproduce six unsupervised general ReID methods on the mentioned datasets. These methods include SpCL [12], ICE [1], CC [9], PPLR [7], Purification [25], and DCCC [19]. As shown in Table 2, Purification [25] achieves superior overall performance among the existing methods. Notably, DRFM significantly surpasses all these unsupervised methods. Specifically, compared to Purification [25], DRFM achieves improvements of 1.2%, 8.5%, and 6.8% in the Rank-1 accuracy on CAVIAR, MLR-CUHK03, and MLR-Market-1501, respectively. The acquisition of this advantage can be attributed to three main factors: (1) DRFM employs a pseudo-label generation method that is better suited for cross-resolution scenarios; (2) DRFM attains a potentially resolution-invariant feature representation through feature fusion; (3) the introduction of multiple optimization methods further enhances the advantages of DRFM compared to existing approaches.

Additionally, we compare DRFM with existing supervised CR-ReID methods, including SING [22], CSR-GAN [40], CAD-Net [29], INTACT [6], PRI [16], PS-HRNet [48], JBIM [52], and LRAR [41]. Encouragingly, despite exhibiting some performance gaps compared to advanced supervised methods (such as JBIM [52] and LRAR [41]), DRFM demonstrates competitive performance when compared to the early methods (such as SING [22] and CSR-GAN [40]) on MLR-Market-1501. This not only validates the superiority of DRFM but also underscores the research potential of unsupervised CR-ReID.

## 4.5 Ablation Study

In this section, we conduct a series of ablation experiments to empirically evaluate the effectiveness of the key components of DRFM.

In Table 3, A1 directly employs the DBSCAN algorithm [11] to generate pseudo-labels between LR and HR images and optimizes using softmax classification loss $L_{sc}$ [30]. A2 first applies super-resolution (SR) to LR images and then utilizes DBSCAN to generate pseudo-labels. A3 utilizes CPG to generate pseudo-labels. A4 introduces $L_{ic}$ based on A3. Following existing feature fusion methods [52], A5 incorporates a fully connected (FC) layer to fuse features from HR and LR spaces. A6 employs the CRFF to generate fusion features. A7 and A8 replace $L_{sc}$ in A2 and A6 with $L_{cc}$, respectively. DRFM introduces $L_{ps}$ based on A8.

*4.5.1 Effectiveness of CPG and $L_{ic}$.* In Table 3, A2 exhibits a slight improvement compared to A1 on all three datasets, indicating the beneficial effect of SR on unsupervised methods. A3 achieves higher performance on all three datasets compared to A1 and A2 by introducing CPG. Specifically, compared to A2, A3 shows improvements of 1.1%, 1.4%, and 2.2% in Rank-1 accuracy on CAVIAR, MLR-CUHK03, and MLR-Market1501, respectively. Furthermore, A4 achieves higher performance by introducing $L_{ic}$ into A3.

To further investigate the effectiveness of CPG and $L_{ic}$ in enhancing the reliability of pseudo-labels, we conduct statistical analysis on the F-score [14] of pseudo-labels generated by A1, A2, A3, and A4

**Table 3: Results of the ablation study where the alternative methods drop or replace alternative components of the proposed DRFM. The main components include CPG, $L_{ic}$, CRFF, $L_{cc}$ and $L_{ps}$.**

| Method | Pseudo-labels | $L_{ic}$ | Fusion | $L_{sc}/L_{cc}$ | $L_{ps}$ | CAVIAR | | MLR-CUHK03 | | MLR-Market-1501 | |
|---|---|---|---|---|---|---|---|---|---|---|---|
| | | | | | | Rank-1 | Rank-5 | Rank-1 | Rank-5 | Rank-1 | Rank-5 |
| A1 | DBSCAN | × | × | $L_{sc}$ | × | 8.3 | 32.1 | 25.3 | 58.2 | 60.3 | 81.7 |
| A2 | SR+ DBSCAN | × | × | $L_{sc}$ | × | 8.5 | 32.3 | 26.8 | 62.6 | 61.7 | 82.9 |
| A3 | CPG | × | × | $L_{sc}$ | × | 9.6 | 33.2 | 28.2 | 65.5 | 63.9 | 85.2 |
| A4 | CPG | ✓ | × | $L_{sc}$ | × | 9.7 | 33.5 | 29.8 | 66.2 | 64.9 | 85.8 |
| A5 | CPG | ✓ | FC | $L_{sc}$ | × | 10.1 | 35.7 | 30.9 | 67.3 | 65.7 | 86.2 |
| A6 | CPG | ✓ | CRFF | $L_{sc}$ | × | 10.6 | 36.2 | 31.9 | 67.8 | 66.8 | 86.5 |
| A7 | SR+ DBSCAN | × | × | $L_{cc}$ | × | 9.5 | 33.4 | 30.2 | 68.3 | 65.0 | 86.2 |
| A8 | CPG | ✓ | CRFF | $L_{cc}$ | × | 11.6 | 37.9 | 33.6 | 70.6 | 68.5 | 87.3 |
| DRFM | CPG | ✓ | CRFF | $L_{cc}$ | ✓ | 12.8 | 39.1 | 35.8 | 72.2 | 69.8 | 88.5 |

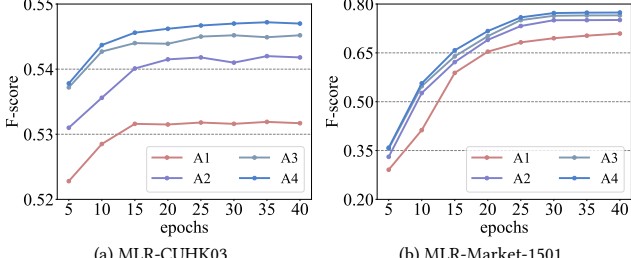

(a) MLR-CUHK03  (b) MLR-Market-1501

**Figure 6: F-score for A1, A2, A3 and A4.**

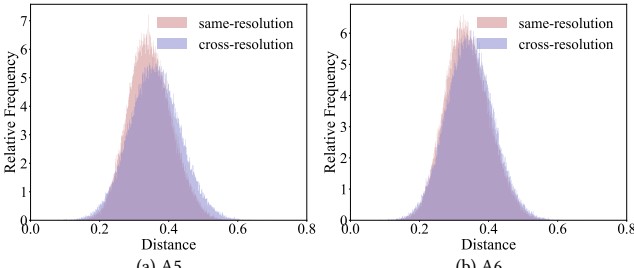

(a) A5  (b) A6

**Figure 7: Feature distance distributions of same-resolution positive pairs and cross-resolution positive pairs for A5 and A6.**

during the training phase. A higher F-score indicates higher accuracy of pseudo-labels. As shown in Figure 6, on MLR-CUHK03 and MLR-Market-1501, A3 consistently exhibits higher F-score throughout the training compared to A2 and A1. This validates that CPG can generate reliable pseudo-labels in cross-resolution scenarios by integrating semantic information at different resolution levels. Moreover, A4 demonstrates higher F-score in the later stages of training compared to A3. This is because $L_{ic}$ focuses on enhancing the model's attention to identity information in SHR images, thereby encouraging potentially high-confidence samples to attain higher $\max(s_i^{lsh})$.

*4.5.2 Effectiveness of CRFF.* As shown in Table 3, A5, which introduces FC on top of A4, demonstrates improved overall performance on all three datasets. This confirms that fusion features are more suitable for cross-resolution scenarios compared to single features. A6, which incorporates CRFF, achieves higher performance than A5. Specifically, compared to A5, A6 shows improvements of 0.5%, 1.0%, and 1.1% in Rank-1 accuracy on CAVIAR, MLR-CUHK03, and MLR-Market1501, respectively. This validates that CRFF can generate fusion features that are more beneficial for performance improvement compared to FC.

To further explore the contribution of CRFF to resolution-invariant feature learning, we visualize the feature distance distributions of same-resolution positive pairs and cross-resolution positive pairs on MLR-CUHK03. As shown in Figure 7, compared to A5, A6 further reduces the difference between the two distributions. This indicates that CRFF is more suitable than FC for generating resolution-invariant features.

*4.5.3 Effectiveness of $L_{cc}$ and $L_{ps}$.* Benefiting from the cross-resolution contrastive loss $L_{cc}$, we observe that A7 significantly outperforms A2, and A8 outperforms A6, validating that $L_{cc}$ can further enhance the model's performance in cross-resolution scenarios compared to $L_{sc}$. We find that DRFM, when introducing the probability sharpening loss $L_{ps}$ on top of A8, further improves model performance, demonstrating the feasibility of leveraging ambiguous samples for optimization. Furthermore, DRFM outperforms all the relevant methods mentioned above, confirming the effectiveness of the combination of all components in DRFM.

## 5 Conclusion

In this paper, we focus on a novel problem: unsupervised cross-resolution person re-identification, and propose the dual-resolution fusion modeling (DRFM) framework to address this problem. In DRFM, we introduce cross-resolution pseudo-label generation (CPG), cross-resolution feature fusion (CRFF), and multiple optimization methods. We conduct an ablation study on these components, validating that CPG can generate reliable pseudo-labels in cross-resolution scenarios, while CRFF and multiple optimization methods effectively facilitate resolution-invariant feature learning, thereby enhancing model performance. Extensive experiments conducted on three datasets demonstrate the effectiveness and superiority of DRFM, which not only outperforms existing unsupervised methods but also exhibits promising performance competitiveness with certain supervised methods.

# Acknowledgments

This work is supported by National Key Research and Development Program of China (Grant no. 2023YFC3305000 and 2023YFC3305003) and the National Natural Science Foundation of China (NSFC, Grant no. 62231013 and 62131004).

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
