# OpenReview forum: "Dual-Resolution Fusion Modeling for Unsupervised Cross-Resolution Person Re-Identification"
_acmmm.org/ACMMM/2024/Conference — MM2024 Oral_

### Official Review · Reviewer_mSG5 · 2024-05-12

**Rating:** 6
**Confidence:** 4

**Summary:**

This paper proposes a dual-resolution fusion modeling (DRFM) framework to address the unsupervised cross-resolution person re-identification problem. DRFM comprises cross-resolution pseudo-label generation, cross-resolution feature fusion, and multiple optimization methods. To validate the effectiveness of the proposed method, the authors conducted extensive experiments on three datasets.

**Strengths:**

1. The manuscript makes efforts to address a new problem (unsupervised CR-ReID) and proposes a novel solution. Overall, this work is sufficiently novel and contributes to the ReID field.
2. The description and logic of the manuscript are clear, and the symbol table used effectively explains the key notations in the text.
3. The proposed method is theoretically reliable, and cross-resolution feature fusion is a suitable technique.
4. The manuscript achieves superior performance on multiple datasets, and ablation study confirms the effectiveness of the method.

**Limitations:**

1. The authors introduce multiple memories to store features; the purpose of doing so should be explained.
2. Although the proposed method achieves some advantages over unsupervised methods on the CAVIAR dataset, it exhibits significant gaps compared to supervised methods. Explaining the reasons for this phenomenon would be helpful for subsequent researchers to further their work.

**Suitability:**

3

---

### Official Review · Reviewer_dpDb · 2024-05-14

**Rating:** 5
**Confidence:** 4

**Summary:**

This paper focuses on unsupervised cross-resolution person re-identification, which faces the challenges of obtaining reliable identity pseudo-labels and resolution-invariant features. To address these challenges, the paper proposes the dual-resolution fusion modeling (DRFM) framework, which includes cross-resolution pseudo-label generation (CPG), cross-resolution feature fusion (CRFF), cross-resolution contrastive loss, and probability sharpening loss.

**Strengths:**

1.This paper is the first attempt in the field of unsupervised cross-resolution person re-identification.
2.The CPG effectively generates reliable pseudo-labels by integrating class vectors from both LR and SHR images.
3.The probability sharpening loss effectively mitigates the predicted probabilities of ambiguous samples in the negative classes.
4.The Experiments are adequate. Contrasting experiments and ablation experiments demonstrate the validity of the authors' method.
5.Figure 6 demonstrates that cross-resolution pseudo-label generation can obtain reliable identity pseudo-labels.

**Limitations:**

1.This paper claim that their method achieves competitive performance with supervised methods, but in reality, there are still significant gap compared to fully supervised methods.
2.The introduction of too many hyperparameters may reduce the reproducibility of the paper.
3.The colors of the lines in Figure 6 are too similar, making them difficult to distinguish. The differences in feature distance distributions in Figure 7 are not significant.

**Suitability:**

3

---

### Official Review · Reviewer_rwGc · 2024-05-23

**Rating:** 3
**Confidence:** 3

**Summary:**

This paper proposes a dual-resolution fusion modeling (DRFM) for cross resolution person reid task. The proposed method consists of a cross-resolution pseudo-label generation (CPG) method to cluster high and low resolution images to generate reliable pseudo labels. And a cross-resolution feature fusion module is designed to fuse features from both resolution. A cross-resolution contrastive loss is proposed for training. Experimental result on 3 datasets show reasonable performance.

**Strengths:**

This paper focus on cross-resolution unsupervised person reid, which gets less attention in research field.
The proposed method consists of cross-resolution pseudo-label generation to jointly consider high and low resolution images to get more reliable labels.
A cross-resolution contrastive loss and probability sharpening loss are proposed for training.
Experimental result on 3 datasets show reasonable performance.

**Limitations:**

Is the comparison methods in table 2 also trained with the multi-resolution data, e.g., original data, super-resolutioned data and downsample datas.
The author should add some explaination about why the ic loss in Eq.18 could encourage the HR encoder to focus on identity-relevant in formation in SHR images rather than artifacts.
There are too many abbreviation which is not necessary, increase the reading difficulty.

**Suitability:**

2

---

### Official Review · Reviewer_hdQF · 2024-05-31

**Rating:** 4
**Confidence:** 4

**Summary:**

This paper focuses on a new task, i.e., unsupervised cross-resolution person reid. A dual-resolution fusion framework is proposed in this paper. Specificially, a cross-resolution pseudo label generation method is proposed to assign pseudo labels for images in different resolution. A cross-resolution feature fusion method is proposed to fuse features in LR and HR. Outliers are also used in training with the proposed probability sharpening loss.

**Strengths:**

+ This paper is an early effort for the unsupervised cross-resolution person reid task.
+ Experimental results show that this paper chieves promising performance compared with outhers.
+ Generating pseudo labels on HR images first and assign pseudo labels to LR images is reasonable and also works.

**Limitations:**

-The proposed cross-resolution feature fusion is similar with graph attention or self-attentinon networks. More discussion and explaination are expected on it.
- It's not appropriate to claim that "achieves competitive performance with supervised methods" as Table2 shows that most supervsied methods outperform the proposed method by a large margin.
- In testing phase, why are LR images used as query images only? How are features extracted for testing with $E_h$ and $E_l$?
- Eq. 8 might be incorrect as L1 norm of a vector is a number.
- The description "all resolution centroids in its cluster" is not clear. Why does a cluster have more than two centroids, i.e., one for HR and one for LR?
- More explaination on "Resolution invariant feature learning" is expected. What's it motivation and why does it works?

**Suitability:**

3

---

### Meta-Review · Area_Chair_6ZJD · 2024-07-01

**Recommendation:** Accept (Oral)
**Confidence:** 5

**Metareview:**

The paper initially received the following ratings: BA(hdQF),	BR(rwGc), WA(dpDb), A(mSG5). After rebuttal and discussion, the final ratings were: WA(hdQF), BA(rwGc), A(dpDb), A(mSG5).

Below are the details of the rebuttal:
1. After reading the rebuttal, all the Reviewer rwGc changed the original borderline rejection recommendation to a borderline acceptance recommendation.
2. All the reviewers were satisfied with the response and gave an acceptance recommendation for this paper.

The AC carefully reviewed the rebuttal:
1. As for the contribution of task setting, the AC agrees that this work makes efforts to address a new problem (unsupervised CR-ReID) and proposes a novel solution, which can serve as a good starting point for exploring effective unsupervised CR-ReID.
2. As for the contribution of technique, a) the proposed CPG effectively generates reliable pseudo-labels by integrating class vectors from both LR and SHR images, which is somewhat new and effective. b) the proposed probability sharpening loss effectively mitigates the predicted probabilities of ambiguous samples in the negative classes, which is well-motivated.
3. This work also provides meaningful visualization and valuable insights based on the experimental results.

In sum, after the rebuttal, the AC believes the concerns raised by the reviewers were well-addressed and recommends acceptance of this paper. In the final version, the AC strongly encourages the authors to include all discussions and reference literature about LReID from the rebuttal and improve the presentation to enhance readability for a diverse audience.